# Phylogenetic Analysis Reveals Source Attribution Patterns for *Campylobacter* spp. in Tennessee and Pennsylvania

**DOI:** 10.3390/microorganisms9112300

**Published:** 2021-11-05

**Authors:** Lauren K. Hudson, William E. Andershock, Runan Yan, Mugdha Golwalkar, Nkuchia M. M’ikanatha, Irving Nachamkin, Linda S. Thomas, Christina Moore, Xiaorong Qian, Richard Steece, Katie N. Garman, John R. Dunn, Jasna Kovac, Thomas G. Denes

**Affiliations:** 1Department of Food Science, University of Tennessee, Knoxville, TN 37996, USA; lkhudson@utk.edu; 2Department of Public Health, University of Tennessee, Knoxville, TN 37996, USA; wandersh@vols.utk.edu; 3Department of Food Science, The Pennsylvania State University, University Park, PA 16802, USA; ruy92@psu.edu (R.Y.); jzk303@psu.edu (J.K.); 4Tennessee Department of Health, Nashville, TN 37243, USA; mugdha.golwalkar@tn.gov (M.G.); katie.garman@tn.gov (K.N.G.); john.dunn@tn.gov (J.R.D.); 5Pennsylvania Department of Health, Harrisburg, PA 17120, USA; nmikanatha@pa.gov; 6Department of Pathology and Laboratory Medicine, Perelman School of Medicine, The University of Pennsylvania, Philadelphia, PA 19104, USA; nachamki@pennmedicine.upenn.edu; 7Division of Laboratory Services, Tennessee Department of Health, Nashville, TN 37216, USA; linda.thomas@tn.gov (L.S.T.); christina.moore@tn.gov (C.M.); xiaorong.qian@tn.gov (X.Q.); Richard.Steece@tn.gov (R.S.)

**Keywords:** *Campylobacter*, *Campylobacter jejuni*, *Campylobacter coli*, source attribution

## Abstract

Campylobacteriosis is the most common bacterial foodborne illness in the United States and is frequently associated with foods of animal origin. The goals of this study were to compare clinical and non-clinical *Campylobacter* populations from Tennessee (TN) and Pennsylvania (PA), use phylogenetic relatedness to assess source attribution patterns, and identify potential outbreak clusters. *Campylobacter* isolates studied (*n* = 3080) included TN clinical isolates collected and sequenced for routine surveillance, PA clinical isolates collected from patients at the University of Pennsylvania Health System facilities, and non-clinical isolates from both states for which sequencing reads were available on NCBI. Phylogenetic analyses were conducted to categorize isolates into species groups and determine the population structure of each species. Most isolates were *C. jejuni* (*n* = 2132, 69.2%) and *C. coli* (*n* = 921, 29.9%), while the remaining were *C. lari* (0.4%), *C. upsaliensis* (0.3%), and *C. fetus* (0.1%). The *C. jejuni* group consisted of three clades; most non-clinical isolates were of poultry (62.7%) or cattle (35.8%) origin, and 59.7 and 16.5% of clinical isolates were in subclades associated with poultry or cattle, respectively. The *C. coli* isolates grouped into two clades; most non-clinical isolates were from poultry (61.2%) or swine (29.0%) sources, and 74.5, 9.2, and 6.1% of clinical isolates were in subclades associated with poultry, cattle, or swine, respectively. Based on genomic similarity, we identified 42 *C. jejuni* and one *C. coli* potential outbreak clusters. The *C. jejuni* clusters contained 188 clinical isolates, 19.6% of the total *C. jejuni* clinical isolates, suggesting that a larger proportion of campylobacteriosis may be associated with outbreaks than previously determined.

## 1. Introduction

Campylobacteriosis, caused by *Campylobacter* spp., is the most common bacterial foodborne illness in the United States, with an annual incidence rate of 19.5 cases per 100,000 in 2019 [1]. The USDA Economic Research Service estimated that up to 1.6 million cases occurred in 2018 in the United States, resulting in a total cost of illness up to $5.2 billion [2]. Campylobacteriosis is common in low- and middle-income countries. Though most people affected fully recover, campylobacteriosis is linked to chronic sequalae including irritable bowel syndrome, reactive arthritis, and Guillain–Barré syndrome (GBS) [3,4].

The *Campylobacter* genus currently consists of 34 validly published species (as of 6 May 2021) [5,6]. *C. jejuni* and *C. coli* are the species that most commonly cause human campylobacteriosis, however, other species including *C. lari*, *C. upsaliensis*, *C. concisus*, *C. ureolyticus*, and *C. fetus* may also cause campylobacteriosis [7,8,9,10]. The *Campylobacter* species cluster into five greater phylogenetic groups, with *C. jejuni*, *C. coli*, and *C. upsaliensis* belonging to the same group [10].

Campylobacteriosis is commonly acquired through the consumption of contaminated food derived from animals, consumption of other food products cross-contaminated from animal products, contact with infected animals, consumption of food products contaminated through soil or water containing animal waste, or consumption of contaminated water [3,11]. Large outbreaks attributed to *Campylobacter* spp. have been associated with drinking water, milk (pasteurized and unpasteurized), lettuce, clams, root vegetables, peas, cheese, beef, chicken, and dogs [12]. However, most *Campylobacter* cases in the United States are considered sporadic (99.5%), rather than outbreak-associated [13]. Additionally, an estimated 18% of campylobacteriosis cases in the U.S. are linked to international travel [3,13,14].

Several studies have investigated source attribution and risk factors for *Campylobacter* infection, although they vary in scope of species, subtyping methodology, and geographical area, and some even incorporate epidemiological data. Overall, considering *Campylobacter* spp. as a whole, most studies suggest that poultry or poultry followed by ruminants are the most prevalent sources [15,16,17,18,19,20,21]. Additionally, Liao et al. found a positive correlation between rurality and ruminant-sourced campylobacteriosis [22]. For *C. jejuni* specifically, the consensus is that poultry or poultry and ruminants are the primary sources, with swine being negligible [15,16,17,20,21,23,24,25,26,27,28,29,30]. Thepault et al. identified 15 host-segregating genes in *C. jejuni*, which were related to metabolic activities, protein modification, signal transduction, stress response, and hypothetical proteins [23]. For *C. coli*, most studies point to poultry as the primary source, but there is considerable variability in the identified secondary source including swine [17,25,27,31,32], ruminants [15,20,33,34], and environmental sources [35]. Traditional subtyping methods (e.g., multilocus sequence typing [MLST], pulsed field gel electrophoresis [PFGE]) that were used in the older source attribution studies have relatively lower discriminatory power compared to whole-genome sequencing (WGS)-based subtyping methods, which also allow for the inference of evolutionary relationships. Applying higher resolution subtyping methods results in more accurate source attribution predictions [24,26,32].

*Campylobacter* is monitored as part of PulseNet [36]. Prior to the PulseNet transition to WGS for subtyping in the surveillance of *Campylobacter*, which was completed in 2019 [37], many states did not sequence this microorganism. As sequencing of *Campylobacter* clinical isolates for routine surveillance has increased [38,39] and PulseNet has transitioned to WGS, there are more opportunities to use these data for cluster identification and source attribution.

The objectives of this study were to compare clinical and non-clinical *Campylobacter* spp. populations from Tennessee (TN) and Pennsylvania (PA); use non-clinical isolate source data and phylogenetic clustering to see if source attribution trends emerge; and identify and describe potential outbreak clusters using hqSNP thresholds. Isolates from two states were examined to provide a greater diversity of both clinical and non-clinical sources. This study provides insight into the population structure of *Campylobacter* genus and the relative importance of different livestock sources in the transmission of clinically important species, *C. jejuni* and *C. coli*.

## 2. Materials and Methods

### 2.1. Isolates and Sequencing

Isolates used in this study (*n* = 3080) are listed in Appendix A and are summarized in Table 1. PA clinical isolates (*n* = 110) were from samples collected from patients at the University of Pennsylvania Health System facilities (2017–2018) and were sequenced for this study. Genomic DNA was extracted using a Qiagen DNA Easy Mini Kit (Qiagen GmbH, Hilden, Germany) as per the manufacturer’s instructions with the addition of a RNase treatment, as previously described [40]. Libraries were prepared using Nextera XT kits and sequenced with an Illumina MiSeq at the University of Tennessee Genomics Core. Biosample IDs and metadata for TN clinical isolates submitted to and sequenced with an Illumina MiSeq by the state public health lab (2013–2021) as part of routine sequencing were obtained from the TN Department of Health in order to retrieve them from NCBI (state information is not listed for clinical isolates on NCBI). The NCBI Pathogen Detection Isolates Browser was queried to obtain IDs for non-clinical isolates using the filters: isolation type “environmental/other” and location “PA” or “TN.” The non-clinical isolate collection date ranges were 2013–2021 for PA and 2003–2021 for TN. Most of the non-clinical isolate sources were food animals and meats including poultry, cattle, swine, sheep (*n* = 7), and goat (*n* = 4); the only non-food animal or meat-associated isolates were isolated from puppy stool (*n* = 2). These isolates will be collectively referred to as “non-clinical isolates.” Raw reads for TN clinical (*n* = 964), TN non-clinical (*n* = 689), and PA non-clinical (*n* = 1317) isolates were downloaded from the NCBI SRA database. All raw reads were trimmed with Trimmomatic (v0.35) [41] (with the following parameters: ILLUMINACLIP: NexteraPE- PE.fa:2:30:10 LEADING:3 TRAILING:3 SLIDINGWINDOW:4:15 MINLEN:36), quality checked with FastQC (v0.11.7) [42] and MultiQC (v1.5) [43]. Trimmed reads were assembled with SPAdes (v3.12.0) [44] (using the careful option and with automatic kmer size selection) and assemblies were filtered to remove contigs <200 bp in length or with SPAdes-computed k-mer coverages <5×. Assembly statistics were generated using QUAST (v4.6.3) [45], Bbmap (v38.88) [46], and SAMtools (v0.1.8) [47]. Only assemblies that met all the following criteria were included in further analyses: total length of 1.4–2.2 Mbp, number of contigs ≤200, G + C content of 29–47%, and estimated average sequencing read coverage of ≥10×.

### 2.2. Phylogenetic Analysis and Taxonomic Identification

KSNP3 (v 3.1) [48] was used to analyze the assemblies, along with reference genomes (*n* = 127 including type strains downloaded from the NCBI RefSeq database), to divide the isolates into presumptive species groups as we did not know the species of some of the isolates initially and wanted to confirm the identity of the rest. Clades or isolates that did not fall within the main clade and/or grouped with reference genomes of known species were removed from the isolate set and KSNP3 was run again with only the isolates from the “main clade.” This process was continued until isolates could be confidently separated into species groups. Optimum kmer value was determined for each run using the included Kchooser utility; optimum values determined and used for analyses were 19 and 21. All isolates were subtyped using the appropriate MLST schemes on PubMLST [49,50,51]. Isolates that needed further classification or taxonomic identification (e.g., non-jejuni/coli, outliers) were evaluated using rMLST [49,52], JSpeciesWS (v3.8.2) [53], SpeciesFinder 2.0 [54], KmerFinder (v3.0.2) [54], and TYGS (v281) [55].

After grouping isolates by species, analyses were conducted with each species individually using KSNP3. This enabled us to determine species-level population structure and identify how closely related clinical isolates were to each other and to non-clinical isolates. The output core SNP matrix fasta files were used to create phylogenetic trees with MegaX v10.1.8 [56] (with evolutionary distances calculated using the number of differences method [57], evolutionary history inferred using the neighbor-joining method [58], and 100 bootstrap replicates [59]), which were then visualized and edited with iTOL v6.1.1 [60].

### 2.3. Cluster Identification

From the species-level KSNP3 analyses outputs, potential outbreak clusters that contained ≥3 clinical isolates that had SNP distances of ≤5 SNPs were identified. Additionally, we identified non-clinical isolates that were related to clinical isolates in these clusters within ≤5 SNPs. Then, a hqSNP analysis was performed with each of the potential clusters individually using the CFSAN SNP pipeline (v1.0.1) [61] and one of the clinical isolates as an internal reference (one that was part of the “core” of the cluster, meaning it had smaller SNP distances to other clinical cluster isolates). The hqSNP distance threshold used to define a cluster was ≤10 hqSNPs. Based on the resulting hqSNP distances, a putative cluster was confirmed as a cluster if all clinical isolates were ≤10 hqSNPs from another clinical isolate. Conversely, a putative cluster was determined to not be a cluster if all the isolates exceeded the hqSNP distance threshold. A putative cluster was marked as needing to be further modified if either some isolates exceeded the hqSNP distance threshold of ≤10 hqSNPs and needed to be removed (singletons), or if the cluster needed to be subdivided into two or more separate clusters. The process was repeated for the modified clusters until they were confirmed as a cluster or determined to not be a cluster. Similarly, non-clinical isolates that were related to clinical within ≤10 SNPs remained in the clusters. Metrics are available in Appendix A.

## 3. Results

### 3.1. Most Isolates Were Identified as C. jejuni and C. coli

Our analysis included clinical and non-clinical *Campylobacter* spp. isolates from both Tennessee and Pennsylvania. The clinical isolates from TN were collected as part of routine state-wide surveillance, whereas the isolates from PA were collected from patients from facilities within a single health system in the Philadelphia and surrounding areas (PA does not currently collect and sequence *Campylobacter* for surveillance). Hence, it should be noted that the PA clinical isolates may not be representative of the entire state. Most of the isolates were determined to be *C. jejuni* (69.2%) and *C. coli* (29.9%), with the remaining isolates belonging to *C. lari* (0.4%), *C. upsaliensis* (0.3%), and *C. fetus* (0.1%) (Table 1). Phylogenetic analyses were conducted separately for each species to identify the species-level population structure of *Campylobacter* from clinical and non-clinical sources in TN and PA. Additionally, this approach allowed us to determine how closely related clinical strains are to each other and to isolates from non-clinical sources.

### 3.2. C. jejuni Formed Three Clades, Primarily Consisting of Poultry and Bovine Isolates

The majority of isolates, 69.2%, were presumptively identified as *C. jejuni* (*n* = 2132) (Table 1), with an overall average distance of 511.2 core SNPs. Considering all sources, 58.4% originated from TN and 41.6% from PA. Isolates from both states showed similar levels of diversity, with average within group distances of 506.3–507.1 core SNPs. These were further divided into three distinct clades (Figure 1). This species phylogenetic group consisted of 45.0% clinical isolates and 55.0% non-clinical isolates. Of the non-clinical isolates, most were poultry (62.7%) and cattle-associated (35.8%), with very few associated with swine (1.3%) or other sources (one isolated from puppy stool and two isolated from sheep/lamb) (Table 1 and Figure 1). Of isolates from the most substantial sources, clinical isolates were the most diverse with an average within group distance of 536.6 core SNPs compared to 488.0 for cattle and 407.9 for poultry. *C. jejuni* clade 1 contained 251 isolates, clade 2 contained 41, and clade 3 contained 1840 isolates, with an average within clade distances of 508.7, 27.2, and 422.6 core SNPs, respectively. The majority of non-clinical isolates from clade 1 originated from cattle (48.8%) or poultry (47.6%) sources, and very few from swine (3.7%). In clade 3, most non-clinical isolates were poultry-associated (65.0%), followed by cattle (33.8%), with a small number from swine (0.8%); clade 3 also contained isolates from puppy stool and sheep/lamb. Within clades 1 and 3, there were evident subclades consisting of non-clinical isolates predominantly from a single food animal source. Out of 11 source-specific subclades, five contained mostly poultry-sourced isolates and six contained mostly cattle-sourced isolates (Figure 1 and Appendix A); the non-clinical isolates within these subclades were 85.7–100.0% poultry- or 88.5–100.0% cattle-sourced, respectively. They also contained clinical isolates: 573 total in the poultry-associated subclades and 158 total in the cattle-associated.

*C. jejuni* species phylogenetic group isolates belonged to 28 different MLST clonal complexes (CC) (148 isolates belonged to STs that are categorized into a CC and 100 isolates are of an unknown ST) (Table 1). Among these, isolates from clade 1 one belonged to eight different CCs and isolates from clade 3 belonged to 21 different CCs. In clade 1, ST-41 and ST-682 CCs contained only clinical isolates and ST-177 CC contained only non-clinical isolates (Appendix A). The non-clinical isolates belonging to ST-177, ST-179, and ST-508 CCs were exclusively from cattle sources, isolates from ST-403 CC were predominantly from cattle (88.9%), those from the ST-283 complex were predominantly of poultry origin (80.0%), and isolates from ST-45 CC were similarly distributed between poultry- and cattle-associated (51.7% and 44.8%, respectively). A small percentage (≤11.1%) of non-clinical isolates belonging to ST-45 and ST-403 CCs were of swine origin.

In *C. jejuni* clade 3, ST-692 and ST-1332 CCs contained only clinical isolates and ST-446 and ST-1287 CCs contained only non-clinical isolates. Non-clinical isolates that belonged to ST-443, ST-446, ST-460, ST-607, and ST-1287 CCs were exclusively poultry-associated. Only considering non-clinical isolates, those from ST-48, ST-49, ST-206, ST-353, ST-354, ST-464, and ST-52 CCs were predominantly poultry-associated (77.8–96.3%), those from ST-42 and ST-61 CCs were predominantly cattle-associated (96.6% and 91.7%, respectively), and isolates from ST-21, ST-22, ST-257, and ST-658 CCs were predominantly poultry- or cattle-associated. A small percentage (≤6.5%) of non-clinical isolates belonging to ST-21, ST-49, ST-61, ST-257, ST-353, and ST-464 CCs were swine-associated.

### 3.3. C. coli Formed Two Clades, Primarily Consisting of Poultry and Swine Isolates

Comprising the second largest group (29.9%), 921 isolates were presumptively identified as *C. coli* (Table 1). A large majority (89.4%) were non-clinical isolates. Isolates of this species were divided into two distinct clades (1 and 3; clade 3 isolates were removed from the final SNP analysis due to large SNP distances [3390–3669 SNPs] between isolates from clades 1 and 3), with clade 1 being further subdivided into three subclades (Figure 2). Clades 1a, 1b, and 3 corresponded to clades identified by other researchers [33,62,63,64], hence they were named consistently with the published literature. Specifically, clade 1a represented isolates belonging to ST-828 CC (80 belonged to a ST that was not associated with a CC or were of an unknown ST), clade 1b represented isolates from ST-1150 CC (seven were of an unknown ST), and clade 1c represented ST-1243 isolates that do not formally belong to any CC. Clade 1 had an overall average SNP distance of 719.6. Clade 1 contained 918 isolates (subclade 1a contained 899, 1b contained 16, and 1c contained three isolates, with within subclade average distances of 659.5, 933.6, and 183.3 core SNPs, respectively) and clade 3 contained five isolates. Similar to other studies, most of the *C. coli* isolates belonged to ST-828 CC [32]. Within clade 1, isolates from cattle sources showed the least diversity (average within group distance of 491.2 core SNPs) when compared to isolates from swine (627.1), clinical (664.2), and poultry (694.75) sources. Additionally, when considering all sources of clade 1 isolates, isolates from PA were more diverse than those from TN, with average within group distances of 743.8 and 660.4 core SNPs, respectively. The majority of non-clinical isolates in clade 1a originated from poultry sources (60.4%), followed by swine (29.7%) and cattle (8.7%). Within clade 1a, there were subclades that consisted of non-clinical isolates predominantly from a single food animal source (poultry, cattle, or swine) (Figure 2 and Appendix A). These source-associated clades also contained clinical isolates: 73 total in the poultry-associated subclades, six in the swine-associated, and nine in the cattle-associated. Clade 1b contained only one clinical isolate and most of the non-clinical isolates were from poultry (93.3%). Clade 1c contained two clinical isolates and one poultry isolate.

Clade 3 consisted of only isolates from TN, with two from clinical sources and three from poultry sources. These clade 3 isolates were confirmed to be *C. coli* by SpeciesFinder, KmerFinder, and rMLST results (all showed 100% support except one that showed 80%) (Appendix A). However, TYGS results indicated that they were a potential new species and ANI values when compared to *C. coli* type strain assemblies were <95%. Additionally, their genomes were 1.54–1.59 Mbp in length and had a G + C content of 31.7–31.8%, compared to 1.91–1.94 Mbp and 31.0% for the *C. coli* type strain assemblies.

### 3.4. Other Species

Thirteen isolates were presumptively identified as *C. lari* (Table 1). Most of these isolates originated from PA (76.9%), and those from non-clinical sources were poultry- (60%) or cattle-associated (40%). Nine belong to *C. lari* ST-21, one to ST-77, and three were of unknown ST (Appendix A). Ten isolates were presumptively identified as *C. upsaliensis* (Table 1), all of unknown ST. All but one of these isolates originated from TN and clinical sources. Four isolates, all from TN clinical sources, were presumptively identified as *C. fetus* (Table 1). Two belonged to *C. fetus* ST-3, one to ST-6, and one to ST-11.

### 3.5. Potential Outbreak Clusters

For *C. jejuni*, we identified 42 potential outbreak clusters using a distance threshold of 10 hqSNPs, consisting of 188 total clinical isolates (Table 2, Appendix A, Appendix A). Each cluster contained 3–13 clinical isolates with average within-cluster distances ranging from 0–11.07 hqSNPs. Thirty-three of the clusters contained only clinical isolates from TN, one only from PA, and eight contained clinical isolates from both states. The collection dates for clinical isolates within a single cluster spanned from a single day up to over four years. Additionally, clinical isolates from 16 of the potential clusters were also related (≤10 hqSNPs) to 1–8 non-clinical isolates per cluster. Among clusters in which clinical isolates were closely related to non-clinical isolates, 11 clusters contained non-clinical isolates of only poultry origin, three of only cattle origin, and two of both poultry and cattle origin.

For *C. coli*, we identified a single potential outbreak cluster using a distance threshold of 10 hqSNPs (Table 2, Appendix A, Appendix A). The cluster contained four clinical isolates, all from TN, with an average within-cluster distance of 13 hqSNPs. The collection dates for the clinical isolates within the cluster spanned over 2.5 years. Clinical isolates from the cluster were also related (≤10 hqSNPs) to seven non-clinical isolates, which were only from poultry sources.

## 4. Discussion

In this study, we examined the population structure of *Campylobacter* spp. isolates from clinical and non-clinical sources from Tennessee and Pennsylvania. We identified livestock source attribution trends for individual *C. jejuni* and *C. coli* clades based on phylogenetic clustering. Additionally, we identified a large number of potential outbreak clusters for *C. jejuni*, which may indicate that more *C. jejuni* illnesses share a common source than previously thought. However, only genomic relatedness was used for cluster identification in this study; epidemiological data are needed to confirm these clusters.

### 4.1. Source Attribution Trends

Previous studies have investigated the relative importance of different sources of human campylobacteriosis [15,16,22,23,24,25]. Most studies included *Campylobacter* spp. together or only *C. jejuni*, and the majority linked campylobacteriosis with poultry, and to a lesser extent with ruminants. The present study examined both *C. jejuni* and *C. coli* individually. Our results indicate that while poultry appears to be the major source of infections with both species based on close phylogenetic clustering, cattle and swine may also be substantial sources of *C. jejuni* and *C. coli* infections.

#### 4.1.1. *C. jejuni* Associated with Poultry and Bovine Sources

The non-clinical isolates from the three major *C. jejuni* clades were mostly from poultry (62.7%) and bovine (35.8%) sources, with clade 1 having approximately equal proportions of cattle (48.8%) and poultry (47.6%) isolates and clade 3 having a higher proportion of poultry isolates (65.0%). Furthermore, we identified several subclades within clades 1 and 3 that contained non-clinical isolates primarily associated with a single animal source, which suggests host adaptation. The poultry- and cattle-associated subclades also contained 573 (59.7% of clinical *C. jejuni* isolates) and 158 (16.5%) clinical isolates, respectively. This indicates that these clinical isolates likely originated from poultry or cattle sources, which further supports the inference that poultry has a relatively higher contribution to *C. jejuni* infection than cattle. When examining clinical and environmental *C. jejuni* isolates collected from east Tennessee during the same time period, Kelley et al. observed that the clinical isolates clustered with isolates from cattle, chickens, water, and other birds [65], which is consistent with our findings. Using three different types of data (MLST, comparative genomic fingerprints, and 15 host segregating genes) for source attribution, Thepault et al. concluded that 31–63% of the British and French clinical cases of *C. jejuni* could be attributed to chicken while 22–55% were attributed to ruminants [24]. The percentage of *C. jejuni* clinical isolates that were in poultry-associated subclades in the current study (59.6%) is at the higher end of the poultry-attributable range they reported. However, the percentage in cattle-associated subclades (16.5%) is lower than the ruminant-attributable range. 

Other researchers have found that some *C. jejuni* clonal complexes (CC) are associated with different livestock or environmental sources, while some are prevalent among multiple sources [25,65,66,67]. Of the 12 most prevalent CCs in the current study (those containing ≥50 isolates), seven were poultry-associated (ST-353, ST-48, ST-206, ST-464, ST-607, ST-443, and ST-52 CCs), two were cattle-associated (ST-61 and ST-42 CCs), and three were associated with both poultry and cattle (ST-21, ST-45, and ST-22 CCs). This was consistent with results based on isolates from Scotland reported by Sheppard et al., with the exceptions of ST-48 (multiple), ST-206 (cattle/sheep), ST-464 (N/A), and ST-52 (multiple) CCs [66]. This was also similar to results based on isolates from east TN described by Kelley et al. with the exception of ST-48 (cattle), ST-206 (N/A), ST-22 (N/A), and ST-52 (N/A) CCs [65]. Some of these differences may be due to geospatial variability in host adaptation. Our results further support the idea that some subpopulations within the greater clades may be host specialist genotypes, while other may be host generalist genotypes. However, CCs that commonly cause campylobacteriosis (e.g., ST-21, ST-45, ST-828 CCs) [25,66] are generalist lineages, which is challenging for genomic source attribution if the CCs alone were to be used for source attribution.

#### 4.1.2. *C. coli* Associated with Poultry, Swine, and Bovine Sources

Overall, the non-clinical *C. coli* strains evaluated in this study were mostly of poultry origin (61.2%), followed by swine (29.0%) and cattle (8.6%). Additionally, we identified subclades within clade 1a that contained non-clinical isolates primarily associated with a single animal source, which suggests host adaptation. The poultry, swine, and cattle-associated subclades also contained 73 (74.5% of *C. coli* clinical isolates), six (6.1%), and nine (9.2%) clinical isolates, respectively, which gives an estimate of the potential relative contribution of each source to *C. coli* infection. Other researchers that examined the epidemiology and source attribution of *C. coli* have concluded that the primary source of human *C. coli* infection is poultry, but their findings differ on whether the secondary source is swine [25] or ruminants [33,64]. 

The Sheppard study cited in the previous section also included *C. coli* isolates; they discovered that the most common CC (ST-828 CC), was prevalent in all studied livestock/environmental sources (chicken, cattle, sheep, pigs, and turkey), but that ST-1150 CC was only found in turkey [66]. Our results differ in that the non-clinical isolates from clade 1b (consistent with ST-1150 CC) originate from chicken (80.0%), turkey (13.3%), and cattle (6.7%). In a different study by Sheppard et al., clade 1 isolates were mostly associated with poultry and ruminants, and isolates from clades 2 and 3 were mostly isolated from riparian sources (water fowl and environmental waters) [64]; our study contained no isolates of riparian origin. Differences between regions such as types of livestock produced, livestock production practices, or wildlife populations may lead to similar strains being prevalent in different livestock reservoirs. While the two regions evaluated in the present study, TN and PA, have similar poultry production inventories, TN has a higher beef cattle inventory, and PA has higher dairy cattle and hog inventories [68,69]. Interestingly, when the *C. jejuni* isolates from non-clinical sources are evaluated separately for each state, poultry and cattle isolates accounted for similar proportions (50.2 and 48.2%, respectively) in PA, but poultry isolates accounted for a much higher proportion (88.5%) than cattle (10.2%) in TN. A similar trend was also seen with *C. coli* with poultry isolates accounting for a higher proportion in TN (70.8%) than PA (55.6%). While these variations may be caused by the aforementioned factors, they could also be a result of different sampling practices between the two states. Cultural and dietary differences may also impact the relative foodborne or direct contact exposures to different livestock sources [70,71,72].

Our source attribution trends were reinforced by case-control studies examining the risk factors of *C. jejuni* and *C. coli* infections. Doorduyn et al. investigated risk factors for *Campylobacter* infections in the Netherlands utilizing questionnaire response and clinical observations to conduct a case-control analysis [73]. The primary risk factor for *C. jejuni* infection was the consumption of chicken (OR = 2.2, 95% CI: 1.5–3.45) with a population attributable risk (PAR) of 28%. Additionally, the calculated adjusted odds ratio (AOR) for both *C. jejuni* and *C. coli* infection was 2.4 (95% CI: 1.2–4.8) for poultry consumption. Furthermore, Rosner et al. determined that consumption of any chicken meat resulted in an AOR of 1.6 (95% CI: 1.2–2.0) and contact with poultry resulted in an AOR of 2.1 (95% CI: 1.4–3.0) for *Campylobacter* infection [25]. They concluded that consumption of pork had an AOR of 3.3 (95% CI: 1.0–11.0) and a population attributable fraction (PAF) of 66% for *C. coli* infections. This is consistent with the higher swine prevalence for *C. coli* in our investigation. Doorduyn et al. also calculated that the OR for beef consumption was 0.5 (95%CI: 0.3–0.9) for *C. coli* infections. This finding is consistent with our source attribution trends, where the proportion of clinical *C. coli* isolates that were part of cattle-associated subclades was low (9.2%) [25]. The findings of these two studies demonstrate that contact and consumption of poultry increases the odds of *Campylobacter* infection with both *C. jejuni* and *C. coli*, and that consuming pork increases the odds of *C. coli* infection. This is consistent with our findings that most of the *C. jejuni* and *C. coli* clinical isolates part of were poultry-associated subclades. However, while a substantial proportion of the non-clinical *C. coli* isolates in the present study were from swine sources, only a small proportion of clinical isolates were part of the swine-associated subclade. 

#### 4.1.3. Other *Campylobacter* spp.

While most isolates in our study were *C. jejuni* and *C. coli*, a small percentage were identified as *C. lari*, *C. upsaliensis*, and *C. fetus*. This is consistent with previous reports that most human campylobacteriosis infections are caused by *C. jejuni* and *C. coli* [7,8,9,10].

### 4.2. Higher Proportion of Clinical Campylobacter Isolates May Be Outbreak-Associated

The current PulseNet guideline for local *Campylobacter* cluster detection is three or more cases differing within 10 alleles by cgMLST, of which two differ within five alleles, and isolation dates for all cases should be within the past 60 days of detection. In this study, potential outbreak clusters were identified using a genetic distance threshold of 10 hqSNPs; note that only genetic relatedness was considered, so epidemiological information would also need to be considered to further support cluster designation or food vehicle. We identified 42 potential outbreak clusters for *C. jejuni* and one for *C. coli*. The *C. jejuni* clusters contained a total of 188 clinical isolates, which was 19.6% of the total *C. jejuni* clinical isolates. This is a much larger proportion of potential outbreak-associated *C. jejuni* than has been previously reported (0.5% of *Campylobacter* infections are estimated to be outbreak-associated) [13]. Other researchers have also found that when using WGS methods, the number of isolates that form clusters is higher than previously thought [74,75,76] including a 25% clustering proportion of Danish clinical isolates reported by Joensen et al. [77]. This may be due to the increased discriminatory power provided by WGS and the resulting analyses compared to previously used methods (e.g., PFGE). As more states begin routinely sequencing *Campylobacter*, there may be an increase in the number of clusters detected, which will likely affect epidemiological investigations and workload. Additionally, these genetic linkages between clinical isolates that were previously thought to be sporadic, especially those that span longer time periods, may also be indicative of continuous common source outbreaks; this information would be useful for both prevention of illnesses (e.g., developing and improving campylobacteriosis prevention strategies) and response to outbreaks (e.g., faster identification of source and implementing corrective actions).

## 5. Conclusions

Notable livestock source attribution trends were observed for *C. jejuni* and *C. coli* clades. These trends were consistent with most of those previously reported, however, they differed from those identified in some studies in different regions. Source trends were even more evident for subclades within the main phylogenetic clades. This suggests potential host adaptation, offers valuable information on the relative contribution of different food animal sources to campylobacteriosis in humans, and may provide insight for epidemiological investigations. Additionally, our findings highlight the need to consider source differences between and within these two species to develop more targeted risk assessments and more effective intervention measures. However, there may be significant regional differences based on geography and dominant types of livestock production. One limitation of this study is the representativeness of the dataset: likely biased sample availability (for both clinical and non-clinical isolates) and lack of isolates from other putative sources. Future investigations comparing more diverse *Campylobacter* populations that include isolates from a larger variety of sources (e.g., wild animals, pets, the environment) may result in different and/or more accurate source attribution trends. Additionally, we found that a fifth of *C. jejuni* isolates were part of potential outbreak clusters, which was much higher than the proportion previously reported to be outbreak-related. This indicates that as WGS is more routinely used for cluster detection, there may be an increase in the number of clusters detected.

## Figures and Tables

**Figure 1 microorganisms-09-02300-f001:**
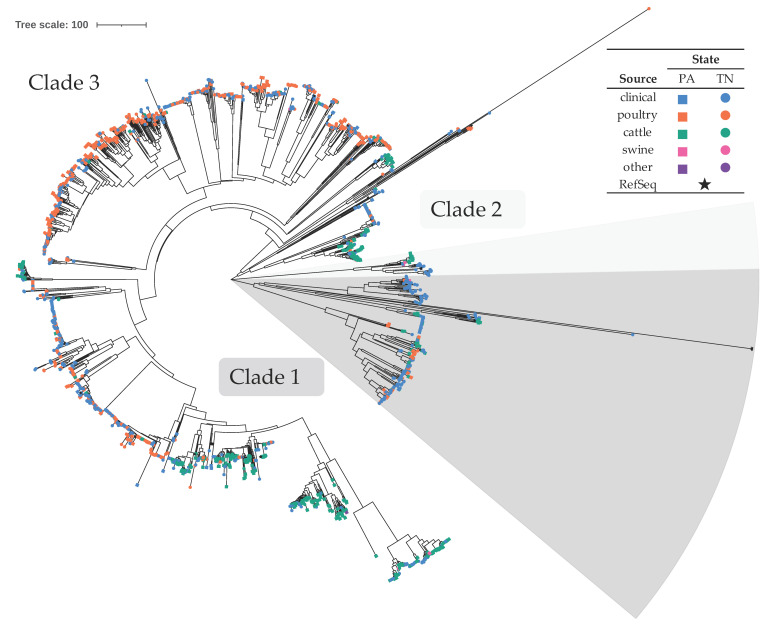
Neighbor-joining phylogenetic tree of *C. jejuni* isolates. The optimal tree with the sum of branch length equal to 37,842.9 is shown. The tree is drawn to scale, with branch lengths representing core SNP distances. There were a total of 6132 core SNP positions in the final dataset. Symbols at the end of tree leaves indicate isolation location and source (see legend). Appendix A is an alternate rendering of the tree with bootstrap values and additional information.

**Figure 2 microorganisms-09-02300-f002:**
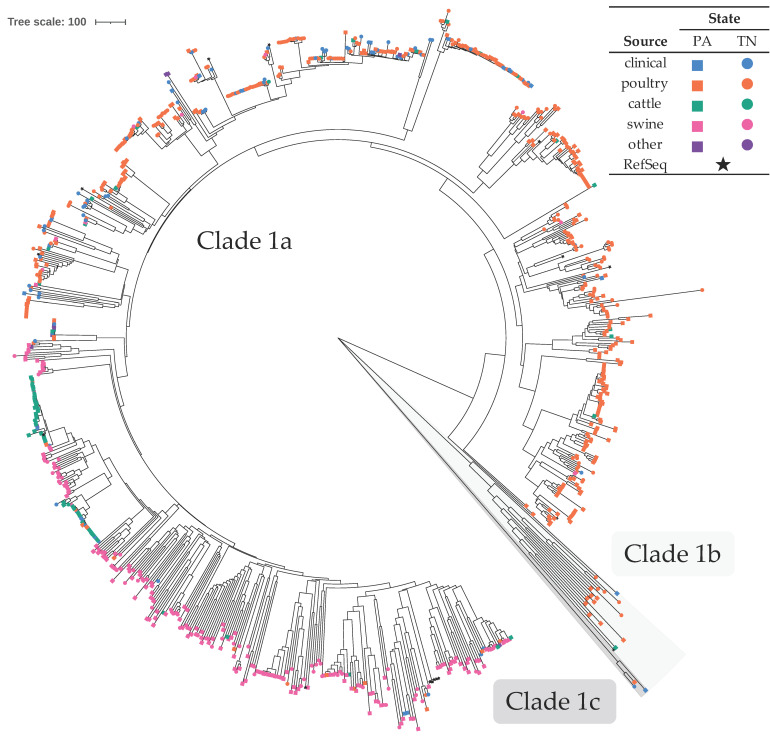
Neighbor-joining phylogenetic tree of *C. coli* isolates. The optimal tree with the sum of branch length equal to 73,485.2 is shown. The tree is drawn to scale, with branch lengths representing core SNP distances. There were a total of 11,628 core SNP positions in the final dataset. Symbols at the end of tree leaves indicate isolation location and source (see legend). Appendix A is an alternate rendering of the tree with bootstrap values and additional information.

**Table 1 microorganisms-09-02300-t001:** Summary of isolates used in the present study and statistics by species and clade.

Species, Clade, Subclade	Clonal Complexes	No. of Isolates (%)								
			State		Source					
			PA	TN	Clinical	Non-Clinical				
							Poultry	Cattle	Swine	Other
** *C. jejuni* **		**2132 (69.2%)**	**887 (41.6%)**	**1245 (58.4%)**	**960 (45.0%)**	**1172 (55.0%)**	**735 (62.7%)**	**419 (35.8%)**	**15 (1.3%)**	**3 (0.3%)**
1	ST-45, ST-403, ST-508, ST-179, ST-283, ST-177, ST-682, ST-41	251	84 (33.5%)	167 (66.5%)	169 (67.3%)	82 (32.%)	39 (47.6%)	40 (48.8%)	3 (3.7%)	0
2	-	41	19 (46.3%)	22 (53.7%)	21 (51.2%)	20 (48.8%)	0	17 (85.0%)	3 (15.0%)	0
3	ST-21, ST-353, ST-48, ST-206, ST-61, ST-42, ST-464, ST-607, ST-22, ST-443, ST-52, ST-257, ST-354, ST-460, ST-49, ST-658, ST-1034, ST-1287, ST-1332, ST-446, ST-692	1840	784 (42.6%)	1056 (57.4%)	770 (41.8%)	1070 (58.2%)	696 (65.0%)	362 (33.8%)	9 (0.8%)	3 (0.3%)
** *C. coli* **		**921 (29.9%)**	**529 (57.4%)**	**392 (42.6%)**	**98 (10.6%)**	**823 (89.4%)**	**504 (61.2%)**	**71 (8.6%)**	**239 (29.0%)**	**9 (1.1%)**
1		916	529 (57.8%)	387 (42.2%)	96 (10.5%)	820 (89.5%)	501 (61.1%)	71 (8.7%)	239 (29.1%)	9 (1.1%)
1a	ST-828	897	516 (57.5%)	381 (42.5%)	93 (10.4%)	804 (89.6%)	486 (60.4%)	70 (8.7%)	239 (29.7%)	9 (1.1%)
1b	ST-1150	16	11 (68.8%)	5 (31.3%)	1 (6.3%)	15 (93.8%)	14 (93.3%)	1 (6.7%)	0	0
1c	-	3	2 (66.7%)	1 (33.3%)	2 (66.7%)	1 (33.3%)	1 (100.0%)	0	0	0
3	-	5	0	5 (100.0%)	2 (40.0%)	3 (60.0%)	3 (100.0%)	0	0	0
** *C. lari* **	ST-21	**13 (0.4%)**	**10 (76.9%)**	**3 (23.1%)**	**3 (23.1%)**	**10 (76.9%)**	**6 (60.0%)**	**4 (40.0%)**	**0**	**0**
** *C. upsaliensis* **	-	**10 (0.3%)**	**1 (10.0%)**	**9 (90.0%)**	**9 (90.0%)**	**1 (10.0%)**	**0**	**0**	**0**	**1 (100.0%)**
**C. fetus**	ST-3, ST-6, ST-11	**4 (0.1%)**	**0**	**4 (100.0%)**	**4 (100.0%)**	**0**	**0**	**0**	**0**	**0**
Total		**3080**	**1427 (46.3%)**	**1653 (53.7%)**	**1074 (34.9%)**	**2006 (65.1%)**	**1244 (62.0%)**	**494 (24.6%)**	**254 (12.7%)**	**10 (0.5%)**

**Table 2 microorganisms-09-02300-t002:** Summary of potential outbreak clusters detected using a hqSNP distance threshold of 10 hqSNPs. Complete data are available in Appendix A.

	No. Potential Clusters			
Species	Total	PA-Only	TN-Only	Related to Non-Clinical Isolates
*C. jejuni*	42	1	33	8
*C. coli*	1	0	1	1

## Data Availability

The data presented in this study are openly available in the NCBI databases under BioProject PRJNA755601.

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
