# Peer review of "Phylogenetic Analysis Reveals Source Attribution Patterns for Campylobacter spp. in Tennessee and Pennsylvania"

_microorganisms, 2021, doi:10.3390/microorganisms9112300_

Round 1
Reviewer 1 Report
The paper propose an interesting study on the use of Phylogenetic Analysis as a system to Reveal Source Attribution Patterns for Campylobacter spp. The paper was clear, materials and methods were complete and well described as results. I consider the paper a good example of study for revealing source attribution in compliance with the most important food safety rules, and I think that the paper can have an important impact on preparedness and management of foodborne outbreaks
Reviewer 2 Report
The manuscript entitled “Phylogenetic Analysis Reveals Source Attribution Patterns for Campylobacter spp. in Tennessee and Pennsylvania” by Hudson et al. has a lot of potential, but there are a number of issues that need to be corrected before publication. The major issue is that the vast majority of the genomes used in this study were downloaded from a public database, however it does not appear that the authors utilized any type of inclusion/exclusion criteria for the genomes such as number of contigs. Those Campylobacter genomes that generated 100s of contigs during assembly typically indicates poor sequencing, issues during the sequencing, or library preparation issues, and these genomes in 100s to 1000s of contigs can cause issues in downstream analysis. If there was some type of inclusion/exclusion criteria for the public genomes then this should be clearly stated in the manuscript. Furthermore, there is major concern that the public genomes were not screened further for completeness and heterogeneity using a program like CheckM. Campylobacter genomes have a very high rate of heterogeneity due to multiple strains or species in single colonies, and there can very often be heterogeneity issues with Campylobacter genomes. In fact, I would suspect that the issues mentioned for Clade 3 of the C. coli genomes that includes 5 genomes are probably due to poor sequencing or heterogeneity issues that should be checked using CheckM. The public genomes need to be screened for this manuscript to make sure that there are no issues and those with poor sequencing (high number of contigs) should probably be eliminated. While there are 1000s of genomes to screen for the manuscript, it should be done to make sure the accuracy of the analysis is correct and not an issue of poor genomes from a public database. Obviously, it is not the authors’ faulty, but it is the fact of dealing with public databases.
Additionally, the following should be addressed or clarified in the manuscript:
- As PA clinical isolates came from a limited source and TN came from statewide surveillance, were the two states sources analyzed separately to determine variation in the sources?
- What is the time range for the isolation of all the strains used in the study? It is mentioned for the clinical isolates, but not the food/environmental, were all the isolates from the same time range?
- Not major, but line 44 should be “low- and middle-income countries” not “undeveloped and developing countries”
- There are numerous grammatical errors and sentences that should be re-phrased throughout the manuscript.
- It is unclear how many clinical strains from TN were sequenced for the study and how many were downloaded from public database. Additionally, the manuscript indicates that 3,096 were downloaded from NCBI SRA database, but then mentions sequencing PA strains. Overall, the paragraph for the isolates and sequencing is very confusing and should be completely re-written for clarification.
- Where did the < 10 hqSNPs parameter for cluster inclusion come from for the analysis? Is there a reference or was this cutoff determined for the study? If so, how was the cutoff determined?
- It would be good to breakdown the clusters by the different states. Particularly since the source of clinical isolates for the two states are different, were there more or less clusters in TN versus PA at least on a percentage of isolates used in the study perspective.
- What were the parameters for calling a clade in the study?
- What were the SNP averages between clinical strains, poultry strains, cattle strains, etc.? It would be good to see the diversity of the strains within each group, and also for between the two states (again particularly because of the difference in sources). What was the SNP average for each jejuni clade?
- Did all the isolates in a clade belong to one of the clonal complexes listed for the clade or were there some strains that did not have a clonal complex within clade 1 and 3?
- It would be beneficial to break down each of the jejuni clades phylogenetically in more detail so that a reader can make more sense of the tree compared to Figure 1. It could be supplemental, but it is very difficult to really see the details for all the strains and make some interpretations of the data.
- Although the authors mention the lack of epi data for the clusters, it would be interesting to look at the time point of isolation of each strain (assuming they have at least that data) to see if the clusters run multiple years or are all within the same year.
- Just a suggestion to improve the manuscript, demonstrating the Campylobacter clusters is a critical component of this manuscript, as it suggests there are a lot more outbreaks than are currently credited for Campylobacter. This should really be analyzed a little deeper and expanded for the manuscript to drive home the point. Currently, it is believed that Campylobacter cases are sporadic, but WGS is demonstrating that is not the case.
